# Tail-tape-fused virion and non-virion RNA polymerases of a thermophilic virus with an extremely long tail

Anastasiia Chaban [1,2,10,13], Leonid Minakhin[3,11,13], Ekaterina Goldobina[1,12], Brain Bae[4], Yue Hao[4], Sergei Borukhov [5], Leena Putzeys[6], Maarten Boon[6], Florian Kabinger [7], Rob Lavigne [6], Kira S. Makarova [8], Eugene V. Koonin [8], Satish K. Nair [4] ✉, Shunsuke Tagami [2] ✉, Konstantin Severinov [3,9] ✉ & Maria L. Sokolova [1,7] ✉

*Thermus thermophilus* bacteriophage P23-45 encodes a giant 5,002-residue tail tape measure protein (TMP) that defines the length of its extraordinarily long tail. Here, we show that the N-terminal portion of P23-45 TMP is an unusual RNA polymerase (RNAP) homologous to cellular RNAPs. The TMP-fused virion RNAP transcribes pre-early phage genes, including a gene that encodes another, non-virion RNAP, that transcribes early and some middle phage genes. We report the crystal structures of both P23-45 RNAPs. The non-virion RNAP has a crab-claw-like architecture. By contrast, the virion RNAP adopts a unique flat structure without a clamp. Structure and sequence comparisons of the P23-45 RNAPs with other RNAPs suggest that, despite the extensive functional differences, the two P23-45 RNAPs originate from an ancient gene duplication in an ancestral phage. Our findings demonstrate striking adaptability of RNAPs that can be attained within a single virus species.

Genes of cellular organisms are transcribed by multisubunit DNA-dependent RNA polymerases (RNAPs), which are conserved in all three domains of life[1–3]. Bacterial RNAPs are composed of two large catalytic subunits β and β′, a dimer of α subunits that serves as an assembly platform for the large subunits, and the ω subunit[4–6]. The active site is located at the interface of two double-psi β-barrel (DPBB) domains formed by the large subunits. The DPBB domain of the β′ subunit contains a conserved amino acid motif DxDxD, where the three aspartates coordinate $Mg^{2+}$ ions required for the catalysis of RNA synthesis. Although more complex, the archaeal and eukaryotic RNAPs contain homologs of each bacterial RNAP subunit, implying that the Last Universal Common Ancestor of cellular organisms (LUCA) already encoded a multisubunit RNAP from which the extant enzymes in all the domains of life are derived[1].

Many bacterial and eukaryotic viruses with double-stranded DNA genomes in the realms *Duplodnaviria* and *Varidnaviria* encode

[1]Center of Life Sciences, Skolkovo Institute of Science and Technology, Moscow 121205, Russia. [2]RIKEN Center for Biosystems Dynamics Research, 1-7-22 Suehiro-cho, Tsurumi-ku, Yokohama 230-0045, Japan. [3]Waksman Institute for Microbiology, Rutgers, The State University of New Jersey, Piscataway, NJ 08854, USA. [4]Department of Biochemistry, University of Illinois at Urbana–Champaign, Urbana, IL 61801, USA. [5]Department of Cell Biology and Neuroscience, Rowan University School of Osteopathic Medicine at Stratford, Stratford, NJ 08084-1489, USA. [6]Department of Biosystems, Laboratory of Gene Technology, KU Leuven, Leuven 3001, Belgium. [7]Department of Molecular Biology, Max Planck Institute for Multidisciplinary Sciences, Göttingen 37077, Germany. [8]National Center for Biotechnology Information, National Library of Medicine, National Institutes of Health, Bethesda, MD 20894, USA. [9]Institute of Molecular Genetics National Kurchatov Center, Moscow 123182, Russia. [10]Present address: Structural and Computational Biology Unit, European Molecular Biology Laboratory (EMBL), Heidelberg 69117, Germany. [11]Present address: Department of Biochemistry and Molecular Biology, Thomas Jefferson University, Philadelphia 19107, USA. [12]Present address: APC Microbiome Ireland, University College Cork, Cork T12 YT20, Ireland. [13]These authors contributed equally: Anastasiia Chaban, Leonid Minakhin. ✉e-mail: snair@illinois.edu; shunsuke.tagami@riken.jp; severik@waksman.rutgers.edu; maria.sokolova@mpinat.mpg.de

homologs of cellular RNAPs (collectively denoted "two-barrel RNAPs")[7–14]. Typically, viral two-barrel RNAPs are organized more simply than their cellular counterparts, lacking some or even all small subunits[10,13,15–17] or consisting of a single polypeptide with two DPBB domains fused[14,18]. The low sequence similarity among the viral two-barrel RNAPs suggests their origin through independent acquisition from cells, followed by simplification of the domain architecture accompanied by extensive divergence. However, the possibility that some single-subunit viral RNAPs predate cellular enzymes, i.e., are relics of ancient RNAPs antedating the LUCA, cannot be discarded. Comprehensive identification and study of highly diverged viral two-barrel RNAPs can be expected to yield insights into the mechanisms and evolution of transcription.

In this work, we characterize unique virion- and non-virion RNAPs of *Thermus thermophilus* bacteriophage P23-45 and compare them to other phage and cellular two-barrel RNAPs. We show that despite the extensive functional differences, the two P23-45 RNAPs originate from an ancient gene duplication in an ancestral phage. These findings provide novel insight into the striking versatility and evolvability of the transcription machinery.

## Results

### A putative RNA polymerase is a part of the tail tape measure protein

Genes of the P23-45 phage were previously divided into three temporal expression classes: early, middle, and late[7]. The middle and late genes are transcribed by the host RNAP modified by the products of two early phage genes[7,19–21]. Transcription of early P23-45 genes is resistant to rifampicin, an inhibitor of bacterial RNAP, and thus was hypothesized to be performed by a phage-encoded enzyme[7]. The product of P23-45 gene *64* (gp64) was proposed as a candidate for this role because of the presence of a metal-binding DxDxD signature motif, even though the overall sequence similarity of gp64 to known RNAPs is extremely low[7]. However, because gp64 is undetectable in P23-45 virions[7], it cannot transcribe early phage genes. To address this conundrum, we searched for the RNAP signature DxDxD in P23-45 virion proteins, under the premise that one of them might be a virion RNAP

injected into the host cell for transcription of early genes. We found that the giant protein gp96 (5,002 residues, 549.6 kDa)[22], which was annotated as a tail tape measure protein (TMP), contains a DFDGD sequence which is conserved in homologous proteins of P23-45 related phages P74-26, G20c, and TSP4 (Fig. 1a). The rest of the gp96 sequence did not show detectable sequence similarity with known RNAPs but contained metallopeptidase and transmembrane domains characteristic of many TMPs (Fig. 1a).

### Validation of the activity of virion and non-virion RNA polymerases

Several N-terminal fragments of gp96 containing the RNAP signature motif were cloned, purified (Fig. 1b), and tested for the ability to extend the RNA component of an RNA-DNA scaffold that mimics the conformation of nucleic acids within the transcription elongation complex of other RNAPs. While the shortest gp96 1050-residue fragment was inactive, 1200-, 1360- and 1540-residue fragments extended the primer (Fig. 1c). Heterologously produced gp64 also extended the RNA component of the scaffold in a template-dependent manner (Fig. 1d). Given that some of the single-subunit RNAPs exist as homodimers in solution[23–25], the oligomeric state of all purified gp96 fragments and gp64 was assessed by mass photometry. All proteins were found to be monomeric in solution (Supplementary Fig. 2).

Western blot revealed that the RNAP domain of gp96 TMP is present in P23-45 virions as part of full-sized gp96, while gp64, in agreement with earlier data[7], was absent (Fig. 1e). Thus, the N-terminal part of the giant gp96 TMP protein appears to be a virion RNAP (vRNAP) that could be responsible for the transcription of the phage early genes. Gp64 is a non-virion RNAP (nvRNAP) that may transcribe phage genes at later stages of infection.

### Both phage RNA polymerases transcribe early genes

The distribution of gp96 and gp64 along the viral genome 5, 20, and 40 min post-infection was monitored using ChIP-seq with appropriate antibodies. Judging by the enrichment of co-immunoprecipitated DNA relative to the total input DNA, both P23-45 RNAPs interact with the phage DNA 5 min post-infection but not at later time points (Fig. 2a,

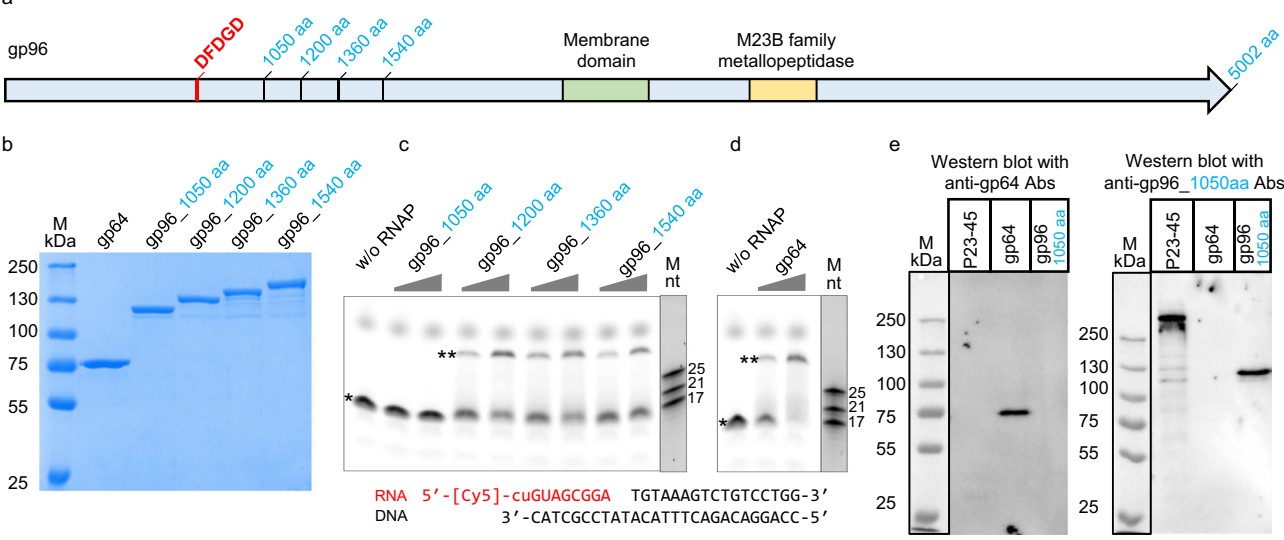

**Fig. 1 | Validation of P23-45 gp96 vRNAP and gp64 nvRNAP. a** Schematic of 5,002-residue tail tape measure protein gp96. The positions of membrane and metallopeptidase domains, the metal binding [779]DFDGD[783] sequence, and C-terminal boundaries of truncated gp96 variants used in this experiment are all indicated. **b** SDS PAGE of purified recombinant gp64 and gp96 variants. **c, d** Extension of a Cy5-labeled RNA primer within the RNA–DNA scaffold shown at the bottom in the presence of rNTPs by the truncated versions of gp96 (**c**) and by gp64 (**d**). Bands corresponding to the initial (10 nt) and extended (29 nt) RNA primer are labeled with one and two asterisks, correspondingly. Micro RNA marker was loaded on the same gels and visualized by staining with SYBR Gold. **e** Western blots of proteins from P23-45 virions with polyclonal antibodies against gp64 and against the 1050-residue fragment of gp96. For each panel shown, the assay was performed twice for each of two biological replicates. Uncropped gels can be found in Supplementary Fig. 1. Source data used in (**b**–**e**) are provided as a Source Data file.

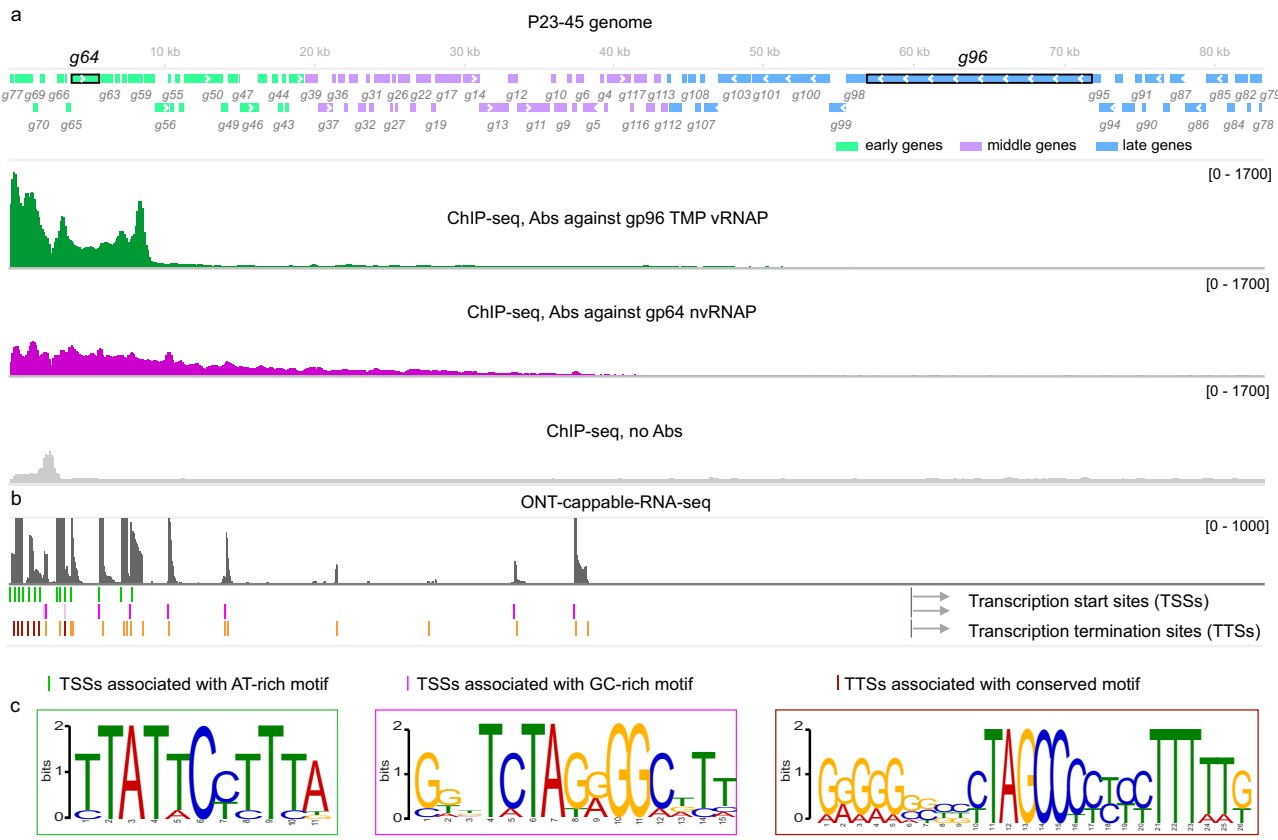

**Fig. 2 | Transcription of the P23-45 genome by gp96 vRNAP and gp64 nvRNAP.**
**a** Schematic of P23-45 genome with early (green), middle (violet) and late (blue) genes is shown at the top. Below, coverage tracks of DNA co-immunoprecipitated from *T. thermophilus* culture 5 min post-infection with P23-45 with antibodies raised against the 1540-residue fragment of gp96 and against gp64 are shown. The control track was obtained using an input DNA without co-immunoprecipitation. **b** ONT-cappable-seq results mapped on the P23-45 genome. Total RNA prepared from infected *T. thermophilus* cells collected 5 min post-infection was used. Transcription start and termination sites are shown below (Supplementary Data 1). **c** Conserved motifs present upstream of indicated groups of transcription start sites (TSSs) and transcription termination sites (TTSs) identified by MEME[51] are shown. The motifs are framed with frame colors corresponding to colors of features in (**b**). Source data used in (**c**) are provided as a Source Data file.

Supplementary Fig. 3). However, the distributions of the two RNAPs along the viral genome were markedly different. Gp96 interacts with a small, 9 kbp region encompassing genes 59-77 located at the left end of the phage genome (Fig. 2a). Previously, these genes, which include the nvRNAP gene *64*, were classified as early[7]. Gp64 is distributed along a much larger, 49 kbp region extending from the left end to the middle part of the genome (genes 1-77) and thus interacts not only with early but also with middle phage genes (Fig. 2a). The enrichment of gp64-co-immunoprecipitated DNA decreases with the increase of the distance from the left end. We conclude that gp96 vRNAP transcribes a subset of early genes immediately adjacent to the left end of the P23-45 genome. These genes likely enter the cell first, and we accordingly rename them "pre-early". The pre-early genes include the gene encoding nvRNAP gp64, which transcribes the early and some middle genes of the phage. Based on partial sensitivity to rifampicin, some of the middle (but not pre-early or early) phage genes are also transcribed by the host RNAP[7].

## Two distinct DNA motifs direct transcription by phage RNA polymerases
Given that two distinct phage RNAPs are involved in the transcription of P23-45 genes, pre-early and early transcripts were expected to be associated with different promoter elements. To delineate phage RNAP promoters, we used ONT-cappable-seq, a specialized long-read RNA sequencing technique that enables the identification of transcription start sites (TSSs)[26]. Using RNA prepared from *T. thermophilus* culture collected 5 min post-infection, we observed 5'

ends of phage transcripts corresponding to 23 distinct TSSs (Fig. 2b). Upstream regions of 14 of these TSSs contained a previously identified strongly conserved AT-rich motif (Fig. 2c, Supplementary Fig. 4a)[7]. Upstream regions of seven TSSs contained a less conserved novel GC-rich motif (Fig. 2c, Supplementary Fig. 4b). Notably, the locations of the AT-rich and GC-rich motifs correspond, respectively, to the distributions of gp96 vRNAP and gp64 nvRNAP along the genome determined by ChIP-seq (Fig. 2a, b). We therefore propose that these motifs define phage promoters; gp96 vRNAP initiates transcription at the AT-rich promoters, whereas gp64 nvRNAP initiates transcription at the GC-rich promoters. This hypothesis is supported by the observation that phages related to P23-45 and encoding RNAP-TMP fusions (P74-26, G20c, and TSP4) also contain a similar conserved AT-rich motif upstream of some of their putative pre-early genes (Supplementary Fig. 4c), whereas P23-45 relatives that lack RNAP fused to TMP (G18 and phiFa) lack this motif (Supplementary Data 2). The phages from the latter group encode a homolog of the gp64 nvRNAP but no counterpart to the vRNAP domain of gp96 and, therefore, must rely on a different strategy to express their pre-early genes.

The ONT-cappable-seq also enabled us to identify 24 transcription terminator sites (TTSs) (Fig. 2b, c). Notably, seven TTSs located close to the left end of the genome encompass a unique 26-nucleotide conserved motif, implying an unusual transcription termination mechanism by gp96 vRNAP (Fig. 2b, c, Supplementary Fig. 4d).

**Structure determination of virion and non-virion RNA polymerases**

As an initial step towards understanding the functional relationships between P23-45 RNAPs and well-characterized single and multisubunit RNAPs, gp96 and gp64 structures were determined. For structural analysis of gp96 vRNAP, the shortest gp96 fragment possessing the RNAP activity was delineated using a fast cell-free assay system (Supplementary Fig. 5)[27]. The shortest fragment of gp96 that retained RNAP activity encompassed residues 327–1124 (Supplementary Fig. 5). We crystallized the 1–1200 and 327–1124 fragments and determined their structures at 3.1 Å and 4.4 Å resolution, respectively (Supplementary Table 1, Supplementary Fig. 6a). The structure of the 1–1200 fragment was largely the same as the structure of the 327–1124 fragment because most of the flanking regions not essential for the activity were disordered, except for two helices at the C-terminus (1125–1162). The crystal structures of gp64 nvRNAP and one of its close homologs (gp62 of phage P74-26) were also determined at 2.7 Å and 2.4 Å resolution, respectively, and found to be essentially identical (Supplementary Table 2, Supplementary Fig. 6b).

**Comparison of P23-45 RNA polymerases with other polymerases**

We compared the structures of P23-45 RNAPs with two other single-subunit two-barrel RNAPs with reported structures, the eukaryotic RNA interference RNAP QDE-1[25] and the crAss-like phage phi14:2 RNAP gp66[18], as well as with the host multisubunit RNAP (Fig. 3).

The overall architectures of gp96 vRNAP and gp64 nvRNAP are markedly different (Fig. 3a). The gp64 nvRNAP structure resembles the structures of other two-barrel RNAPs and has a characteristic crab-claw-like architecture that includes the clamp domain (Fig. 3a). In contrast, gp96 vRNAP contains no defined domain at the corresponding position, resulting in a clamp-less, flat structure that heretofore has not been observed for any RNAP (Fig. 3a). We hypothesize that this unique structure enables the passage of gp96 vRNAP through the narrow channel in the phage tail.

The conserved regions of the compared two-barrel RNAPs are analyzed in Fig. 3b and Supplementary Figs. 7–11. For each of the enzymes, one of the active-site-forming double-psi β-barrel domains, DPBB-A, is well-conserved except for several lineage-specific insertions (Supplementary Fig. 7). In contrast, in the DPBB-B domain, three loops connecting β-strands (β1/β2, β3/β4, and β4/β5) are conserved only between the two P23-45 RNAPs, whereas QDE-1 and phi14:2 gp66 share an extended α-helix (Supplementary Fig. 8). The N-terminus of the duplex-binding helix preceding DPBB-B is extended in QDE-1 and phi14:2 gp66 (Supplementary Fig. 8). The bridge helix and the N-terminal part of the trigger loop are conserved in all analyzed RNAPs (Supplementary Fig. 9).

DPBB-A and DPBB-B are present not only in the analyzed RNAPs but also in another double-barrel polymerase, archaeal DNA polymerase, polD, as previously reported[28–31]. A few small elements are specifically conserved between the DPBB domains in PolD and bacterial multisubunit RNAP, which are absent in the single-subunit RNAPs (Supplementary Figs. 7, 8). No other conserved elements were found between the analyzed RNAPs and PolD.

The clamp domains of P23-45 gp64 nvRNAP, QDE-1, phi14:2 gp66, and *T. thermophilus* RNAP are comprised of three elements: the connector between DPBB-B and DPBB-A, the α-helix/loop following DPBB-A, and the C-terminal region. All compared RNAPs except gp96 vRNAP share conserved parts in the clamp elements, with the highest similarity between QDE-1 and phi14:2 gp66 (Supplementary Fig. 10). Gp96 is exceptional as it lacks the C-terminal region, and its connector element, which is disordered in the gp96 crystals (Fig. 3a, gp96 613–679), is twice shorter than in QDE-1 and phi14:2 (Supplementary Fig. 10). The disordered region is likely positioned close to the RNA-DNA hybrid duplex during transcription elongation (Fig. 4) and might dynamically interact with the nascent RNA strand, moving similarly to the clamp domains of other RNAPs.

The N-terminal domains of the compared single-subunit RNAPs contain a common structural motif composed of one α-helix (α2) and three β-strands (β3–5) (Supplementary Fig. 11). The two P23-45 RNAPs have an additional α-helix (α3) between β3 and β4 absent in other RNAPs. The QDE-1 and phi14:2 share another α-helix (α1) and two β-strands (β1–2) adjacent to the N-terminus of the common motif, which are not found in P23-45 RNAPs. Although the N-terminal domain of the β subunit of *T. thermophilus* RNAP differs substantially, it contains a five-stranded β-sheet (β1–5) with the same strand order and direction as the motif in QDE-1 and phi14:2 gp66 (Supplementary Fig. 11).

Thus, despite the remarkably different overall architectures of the two P23-45 RNAPs, gp96 vRNAP shares more structurally conserved regions with gp64 nvRNAP than with other RNAPs (Fig. 3b, Supplementary Figs. 7–11). Conversely, the two other single-subunit two-barrel RNAPs with reported structures share structurally similar elements that are absent in P23-45 RNAPs and multisubunit RNAPs (Fig. 3b, Supplementary Figs. 7–11).

**Two major clades of two-barrel RNA polymerases**

To further assess the relationships between different families of two-barrel RNAPs, we constructed a structure-based multiple alignment for selected representatives of 23 RNAP families described previously[18,30], using the comparison of available structures as a guide (Supplementary Data 3). Given the lack of significant sequence similarity between many RNAP families beyond the small region around the DxDxD motif, standard, sequence-based phylogenetic methods are not applicable in this case. Therefore, we constructed a similarity dendrogram using HHalign pairwise scores (Fig. 5a). In this dendrogram, the RNAPs split into two major clades. The first clade includes the multisubunit RNAPs conserved between all cellular organisms and some of their derivatives with fused DPBB domains. The second clade consists of diverse phage and plasmid single subunit RNAPs and the eukaryotic RNA interference RNAPs (Fig. 5a). Notably, gp96 vRNAP and gp64 nvRNAP of P23-45 cluster together, in accord with the manual structure comparison that revealed shared structural elements between the two, to the exclusion of other RNAPs. Thus, the vRNAP and nvRNAP of P23-45 likely evolved as a result of an early duplication in an ancestral phage genome.

## Discussion

In this work, we characterized vRNAP and nvRNAP of bacteriophage P23-45. Unusually, the vRNAP is fused to the phage TMP, which is a previously unsuspected association between phage proteins thought to perform unrelated functions. The TMPs of phages with long non-contractile tails are thought to be in a metastable form within the phage tail tube and are injected from the tail into the cell envelope upon infection, forming a channel for the entrance of viral DNA[32,33]. We hypothesize that the vRNAP is associated with the pore formed by the TMP in the bacterial cell envelope and transcribes pre-early phage genes concomitantly with the phage genome translocation through the pore (Fig. 5b). Fusions of vRNAPs to large proteins of unknown functions are found in crAss-like[12] and N4-related[34] phages and might also be required for coupling viral DNA translocation with the transcription of pre-early and/or early phage genes. The mechanistic details of the P23-45 vRNAP packaging, its interaction with the TMP, and pre-early gene transcription remain to be elucidated.

The vRNAP and nvRNAP of P23-45 apparently evolved under different selective pressures for distinct functions, which resulted in major structural and mechanistic differences. Transcription by vRNAP and nvRNAP is directed by dissimilar AT-rich and GC-rich promoters, respectively, and the two phage RNAPs have distinct overall

 

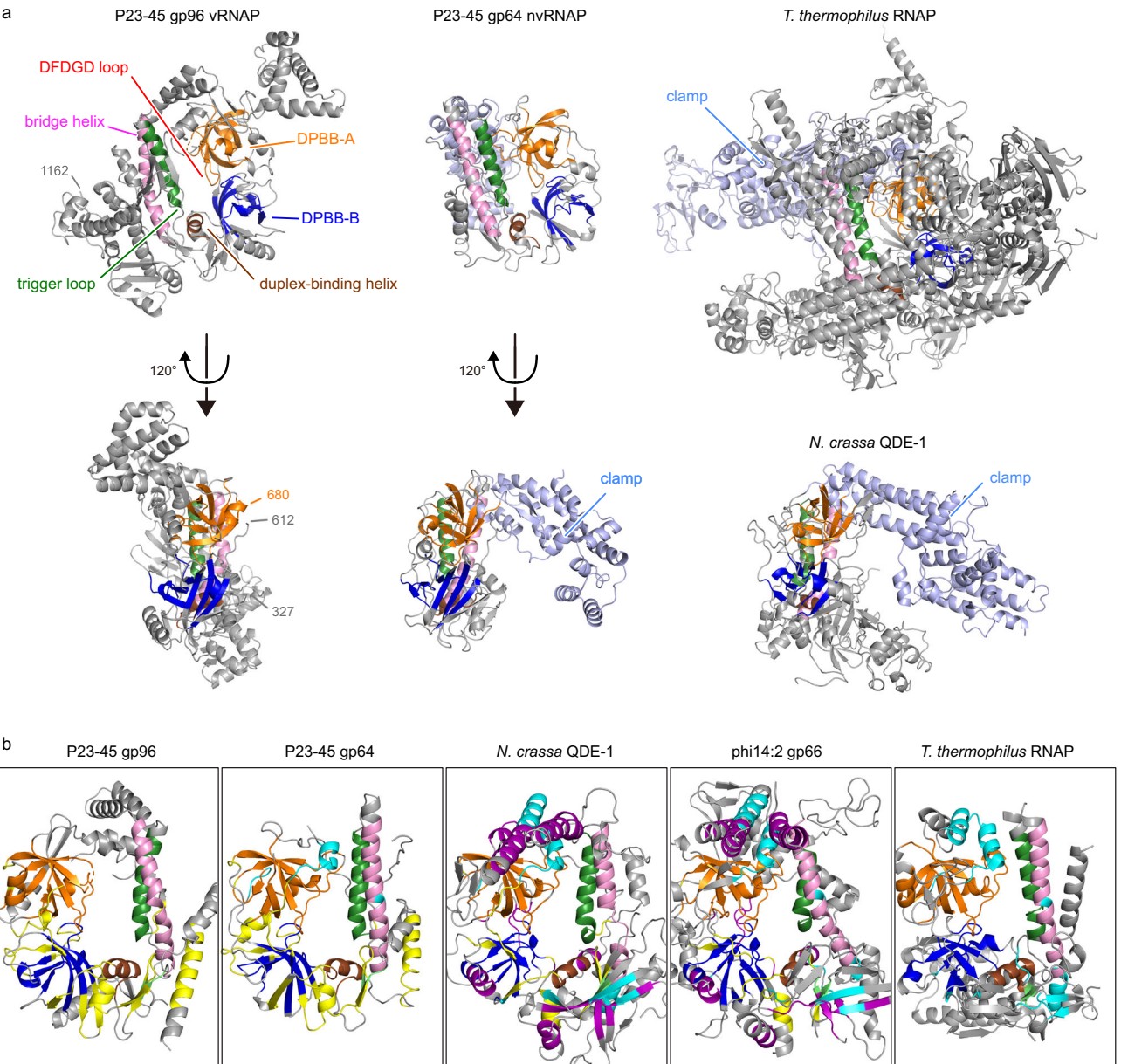

**Fig. 3 | Structures of P23-45 gp96 vRNAP and gp64 nvRNAP. a** Overall structures of P23-45 gp96 vRNAP and gp64 nvRNAP are shown as ribbon models in two orientations and compared with the structures of *T. thermophilus* RNAP (PDB: 2O5J[50]) and *N. crassa* QDE-1 (PDB: 2J7N[25]). The conserved structural elements are colored (DPBB-A−orange; DPBB-B−blue; duplex-binding helix−brown; bridge helix −pink; trigger loop−green). Clamp domains are structurally variable (light blue) and the corresponding part in the gp96 structure (residues 613−679) is disordered. **b** The central regions of P23-45 gp96 vRNAP, gp64 nvRNAP, *N. crassa* QDE-1 (PDB: 2J7N[25]), phi14:2 gp66 (PDB: 6VR4[18]), and *T. thermophilus* RNAP (PDB: 2O5J[50]) are shown in the same orientation. Local structures conserved in some RNAPs are highlighted: those conserved with gp96 but not with *T. thermophilus* RNAP−in yellow; those conserved with *T. thermophilus* RNAP but not with gp96 – in cyan; those conserved only between QDE-1 and gp66 – in purple. Other small elements conserved in all compared RNAPs are in lime.

architectures, with the TMP-fused vRNAP being unusually flat and lacking the clamp domain. Despite these pronounced differences, a comparison of conserved regions of the two-barrel RNAPs suggests that the two P23-45 RNAPs originated from an ancient gene duplication in an ancestral phage. These findings highlight the striking structural and functional malleability of two-barrel RNAPs in phages. A survey of viral genomes for two-barrel RNAP genes showed that many of them encode two RNAPs (Supplementary Data 4). Thus, functional diversification of RNAPs appears to be a common phenomenon that likely reflects distinct structural and functional contexts of expression of different classes of viral genes. Detailed characterization of the viral transcription machineries can be expected to reveal unknown aspects of RNAP functionality and illuminate virus biology.

## Methods

### Bacterial and phage growth conditions

Bacteriophage P23-45 infecting *T. thermophilus* strain HB8 was isolated previously[22]. Bacterial cells were grown in an orbital shaker at 70 °C in liquid medium (0.8% tryptone, 0.4% NaCl, 0.2% yeast extract, 0.5 mM $MgSO_4$, and 0.5 mM $CaCl_2$ in Vittel mineral water). To grow *T. thermophilus* and P23-45 on plates, a solid medium was used (liquid medium supplemented with 2% agar for the bottom layer and 0.4% agar for the top layer). To obtain phage plaques, 100 μL of phage stock were mixed with 1 mL of *T. thermophilus* overnight culture and 5 mL of melted top-agar and spread on a plate with bottom-agar. Plates were incubated overnight at 70 °C. A fresh phage stock was obtained by adding a fresh P23-45 plaque to the growing *T. thermophilus* culture at

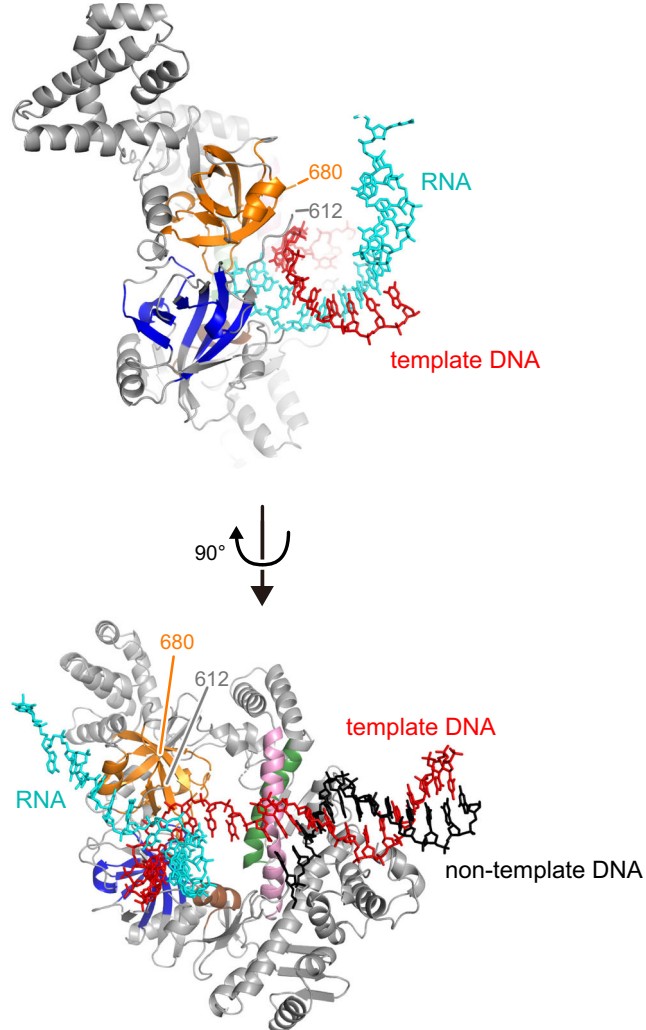

**Fig. 4 | A model of the gp96 RNAP elongation complex.** The gp96 elongation complex was modeled by superimposing gp96 RNAP and the elongation complex of *T. thermophilus* RNAP (PDB: 2O5J[50]). The same color scheme as in Fig. 3 is applied for gp96 RNAP. The nucleic acid models were copied from 2O5J (RNA: cyan; template DNA: red; non-template DNA: black).

an optical density at 600 nm ($OD_{600}$) of around 0.2. The cells were incubated until lysis (~3 h). To remove cell debris, the lysate was centrifuged (4 °C, 12,000 × g, 10 min) and the supernatant was collected. The resulting phage stock (around $10^9$–$10^{10}$ plaque-forming units (PFUs) per mL) was stored at 4 °C.

## Phage purification using CsCl gradient
For phage purification, freshly obtained phage stock was filtered and centrifuged (10 °C, 50,000 × g, 90 min). The pellet was resuspended in SM buffer (50 mM Tris-HCl pH 7.5, 100 mM NaCl, 10 mM $MgSO_4$). Three layers of CsCl gradient (1.45 g/mL, 1.5 g/mL, 1.7 g/mL) were prepared in SM buffer in a centrifugation tube. Phage was loaded on the top of the CsCl gradient and covered until the top of the tube with 0.9% NaCl. The gradient was centrifuged for 90 min at 100,000 × g at 10 °C. After centrifugation, P23-45 virions formed a single white layer, which was taken using a syringe by puncturing the sidewall of the centrifugation tube. Finally, purified phage virions were dialyzed overnight against SM buffer. Purified virions were mixed with an SDS sample-loading buffer, boiled for 5 min, and separated by a Tris–HCl 4%–20% (w/v) gradient SDS-polyacrylamide gel (BioRad) electrophoresis followed by western blotting.

## Phage DNA extraction
P23-45 DNA was purified from the phage lysate by standard phenol/chloroform extraction. A 10 mL aliquot of phage lysate was mixed with PEG solution (30% PEG-8000, 3 M NaCl) at a 2:1 ratio and incubated overnight at 4 °C. PEG-precipitated phage particles were pelleted by centrifugation (4 °C, 10,000 × g, 30 min). The pellet was resuspended in 5 mM $MgSO_4$, and DNase I and RNase A were added to a final concentration of 10 μg/mL each, followed by incubation at 37 °C for 1 h. Then, Proteinase K buffer (SDS to 0.5% final concentration and EDTA pH 8.0 to 20 mM final concentration) and 20 μL of Proteinase K (20 mg/mL stock) were added to 500 μL of the sample. The mixture was incubated for 1 h at 60 °C and then cooled to room temperature. An equal volume of phenol:chloroform (1:1) mixture was added and mixed by inverting the tube several times. The mixture was centrifuged (room temperature, 12,000 × g, 5 min). The supernatant was transferred to a new tube. The step with phenol:chloroform treatment was repeated. Then, an equal volume of chloroform was added, and the mixture was centrifuged (12,000 × g) for 5 min at room temperature. The supernatant was transferred to a new tube, and then 1/10 volume of 3 M ammonium acetate (pH 7.5) and 2.5 volumes of 96% ethanol were added to the sample for overnight incubation at −20 °C. DNA was pelleted by centrifugation (4 °C, 30,000 × g, 20 min). The pellet was washed with 70% ethanol twice. Upon the ethanol removal, the tube was left open on a bench for ~20 min. DNA was dissolved in TE buffer (10 mM Tris-HCl pH 8, 1 mM EDTA).

## Purification of gp64 and gp96 fragments
The gene coding for gp64 nvRNAP (GeneID 5600450 in NCBI Reference Sequence YP_001467917.1) was PCR-amplified from P23-45 genomic DNA and cloned into pET28a between NdeI and XhoI restriction sites. Different fragments of the gene coding for gp96 vRNAP (GeneID 5600432 in NCBI Reference Sequence YP_001467917.1) were PCR-amplified from P23-45 genomic DNA and cloned into pCOLADuet-1 pre-digested by BamHI using NEBuilder HiFi DNA Assembly Master Mix (New England Biolabs). The resulting plasmids were transformed into BL21 Star (DE3) chemically competent *E. coli* cells. For each protein, 1 liter of LB culture was grown at 37 °C to an $OD_{600}$ of 0.6, and recombinant protein overexpression was induced with 1 mM IPTG for 3 h at 37 °C. Cells were harvested by centrifugation (4 °C, 3,700 × g, 20 min), and disrupted by sonication in buffer A (20 mM Tris-HCl pH 8.0, 300 mM NaCl) containing 1x cOmplete Protease Inhibitor Cocktail (Roche) followed by centrifugation at 15,000 × g for 30 min. The lysate was then incubated at 65 °C for 30 min to denature non-thermostable proteins. Denatured proteins were removed by centrifugation (4 °C, 15,000 × g, 30 min), the supernatant was filtered (PES membrane filters with 0.22-μm diameter pores, BIOFIL) and loaded onto a 5 mL HisTrap Sepharose HP column (GE Healthcare) equilibrated with buffer A. The column was washed with buffer A supplemented with 20 mM imidazole, and the protein was eluted with a linear 0–0.5 M imidazole gradient in buffer A. Fractions containing recombinant protein were combined and diluted ~6-fold with buffer B (20 mM Tris-HCl pH 8, 0.5 mM EDTA, 1 mM DTT, 5% glycerol) to a final concentration of 50 mM NaCl, and loaded onto a 5 mL HiTrap Heparin HP Sepharose column (GE Healthcare) equilibrated in buffer B. The protein was eluted using a linear 0–1 M NaCl gradient in buffer B. Fractions containing the target protein were pooled, concentrated (Amicon Ultra-4 Centrifugal Filter Unit with Ultracel-30 membrane, EMD Millipore) to ~5-15 mg/mL, and separated by gel-filtration on a Superdex 200 Increase 10/300 column (GE Healthcare) equilibrated in buffer A. The fractions containing target proteins were pooled and concentrated up to 4 mg/mL, then glycerol was added up to 50% to the sample for storage at −20 °C.

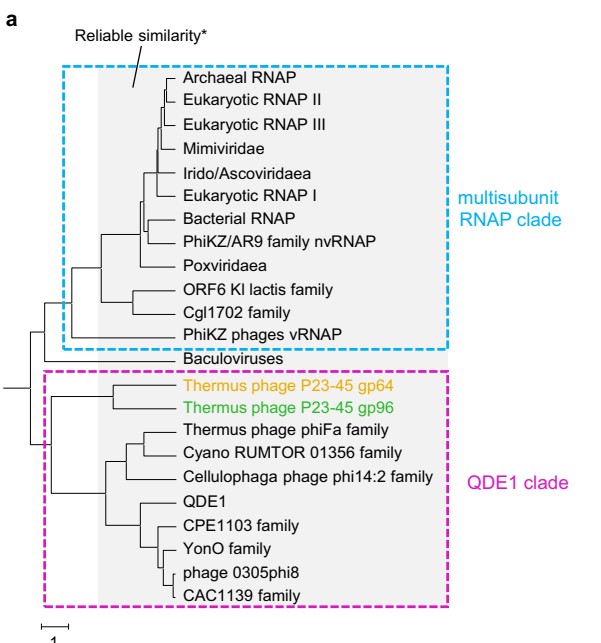

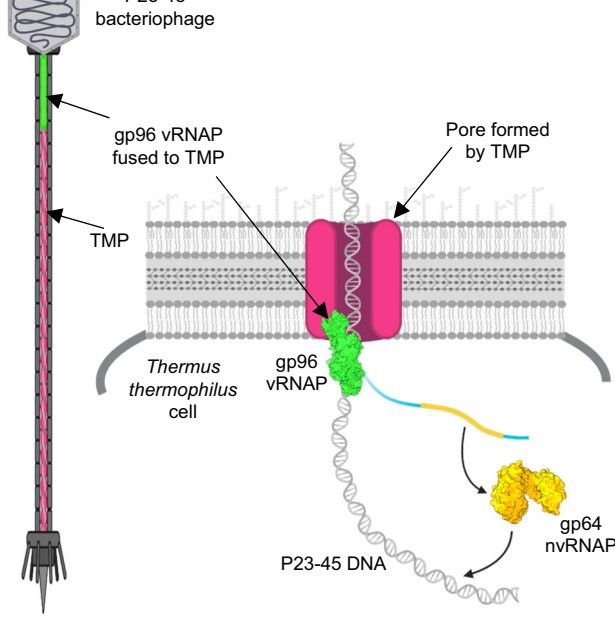

**Fig. 5 | Relationships between RNAP families and functions of P23-45 RNAPs during infection. a** Sequence similarity dendrogram. UPGMA dendrogram was built using HHalign pairwise score matrix for 23 multiple alignments of distinct RNAP families[52]. Alignments are provided in the Supplementary Data 3. *Branches within 3 distance units of the tree depth (see the scale under the tree) usually reliably reflect sequence similarity based on previous observations. Abbreviations: nvRNAP−non-virion RNAP; vRNAP−virion RNAP; QDE1−RNA/DNA-dependent RNA polymerase QDE-1. **b** Proposed mechanism of P23-45 pre-early genes transcription by gp96 vRNAP fused to tail tape measure protein (TMP) concomitantly with phage DNA translocation through the TMP pore. One of pre-early genes encodes gp64 nvRNAP, which transcribes all early and some middle genes. Late genes of the phage are transcribed by host RNAP[7] (not shown). The figure was created using BioRender.com.

## Mass photometry

All mass photometry measurements were performed on a Refeyn TwoMP mass photometer with the Refeyn AcquireMP and DiscoverMP software (both v. 2.2). Data acquisition records were done in normal mode with a regular image size for 60 s per sample. Measurements were carried out in silicone gaskets (3 mm × 1 mm, GBL103250, Grace Bio-Labs) on microscopy slides that had been cleaned by isopropanol, Milli-Q water, and isopropanol. For each measurement, 15 μL of buffer was added to the gasket, and the autofocus function was switched on. After successful focusing, 5 μL of prediluted protein sample were added, and the measurement was started. The resulting movies of 60 s were then analyzed and graphed in the DiscoverMP software. For each measurement, at least 2,000 events were recorded and analyzed. In addition, each protein was measured in triplicates.

To test the oligomeric states of the N-terminal fragments of gp96 and gp64 RNAPs, the proteins were incubated in the transcription buffer (20 mM Tris pH 8, 10 mM MgCl$_2$) at 2 μM concentration for 10 min at 55 °C. The samples were then cooled down to 4 °C and kept on ice. Shortly before the measurements, proteins were diluted to 40 nM, and each sample was applied to the mass photometer as stated above, which took ~30 s. This resulted in a final measurement concentration of 10 nM. All proteins were monomeric in solution under these experimental conditions. However, the existence of quaternary structures at different experimental conditions cannot be excluded.

## Antibody generation and purification

Gp64 and gp96 (1-1050 and 1-1540 fragments) were used for the immunization of rats by Brovko F.A from the Institute of Bioorganic Chemistry RAS. Antisera were tested by western blotting against the proteins used for immunization. Polyclonal antibodies from rat antisera were purified by affinity chromatography. For that, gp64 and gp96 (1-1050 and 1-1540 fragments) were dialyzed overnight against a coupling buffer containing 0.1 M Na$_2$CO$_3$, 0.1 M NaHCO$_3$, pH 8.3, and 0.5 M NaCl.

Cyanogen bromide−activated Sepharose 4B (Sigma-Aldrich) beads were activated with ice-cold 1 mM NaCl and washed thrice with coupling buffer. The maximum available amount of protein (25 mg of 1-1050 gp96, 25 mg of 1-1540 gp96, and 9 mg of gp64) was immobilized on activated beads (2.5 mL, 2.5 mL, and 1.8 mL, respectively) and incubated overnight at 4 °C with rotation. On the following day, the beads were washed two times with coupling buffer and incubated in 15 mL of 0.1 M Tris- HCl pH 8.0 for 2 h at room temperature with rotation. Next, the beads were washed three times in the ice-cold 0.5 M NaCl, 0.1 M Tris-HCl pH 8.0 followed by the ice-cold acetate buffer (0.2 M sodium acetate, 0.5 M NaCl pH 4.0). Next, the beads were washed three times in PBS (2.7 mM KCl, 137 mM NaCl, 1.76 mM potassium phosphate, 10 mM sodium phosphate pH 7.4 adjusted with HCl). Finally, 5 mL of antiserum was diluted up to 28 mL with PBS. Mixtures were incubated overnight at 4 °C with rotation. The next day, antibodies were eluted by 0.2 M glycine pH 2.8. Immediately after elution, Tris-HCl pH 8.8 and KCl were added to the 1 mL aliquots to final concentrations of 8 mM and 0.3 M, respectively. Antibodies were concentrated, then glycerol was added up to 50% to the sample for storage at −20 °C.

## Western blotting assay

Purified virions and recombinant proteins were mixed with an SDS sample-loading buffer, boiled for 5 min, and separated by a Tris−HCl 4%−20% (w/v) gradient SDS-polyacrylamide gel electrophoresis (BioRad). Following electrophoresis, samples were transferred to the nitrocellulose membrane (Bio-Rad). After transfer, the membrane was blocked with PBS containing 5% non-fat milk for 1 h at room temperature. Then, the purified antibodies from −20 °C stock (1:1000 diluted) were added. The next day, the membrane was washed three times with PBS containing TWEEN 20, followed by the addition of anti-rat antibodies (A9542, Sigma, 1:8000 diluted). The membrane was incubated at room temperature for 1 h and again washed three times. The immunoreactive

bands were visualized using an Immun-Star AP Substrate (Bio-Rad).

### In vitro transcription

The scaffold for the RNA extension reaction (RNA primer: 5′ Cy5- CUGU AGCGGA; DNA template: CCAGGACAGACTTTACATATCCGCTAC; DNA non-template: TGTAAAGTCTGTCCTGG) was annealed in water (70 °C for 5 min, −0.1 °C/s to 12 °C). Transcription reactions were performed in a 20 µL reaction of transcription buffer (20 mM Tris-HCl pH 8, 10 mM $MgCl_2$) and contained 200 nM RNA–DNA scaffold and 100 nM or 400 nM of gp64 or gp96 RNAPs. Reactions were incubated for 10 min at 30 °C, followed by the addition of 1 mM each of ATP, CTP, GTP, and UTP. Reactions proceeded for 30 min at 55 °C and were terminated by the addition of an equal volume of denaturing sample loading buffer. The products were resolved by electrophoresis on Novex™ TBE-Urea Gels, 15%. The MicroRNA marker (NEB) was loaded on the same gels and visualized by staining with SYBR Gold. Results were analyzed by Amersham Typhoon Biomolecular Imager (GE Healthcare).

### Sample collection for ChIP-seq

*T. thermophilus* HB8 cells were grown in 800 mL of the liquid medium until $OD_{600}$ of 0.2. The culture was infected with P23-45 at a multiplicity of infection of 10. At various time points (5, 20, and 40 min post-infection), 200-mL aliquots of infected cultures were withdrawn. To cross-link proteins with DNA, formaldehyde was added to final concentration of 1%, immediately after sample collection. After 20 min of incubation with formaldehyde at 37 °C with agitation, the cross-linking reaction was stopped by the addition of glycine to a final concentration of 0.5 M. After 20 min incubation with agitation, cells were harvested by centrifugation at (4 °C, 10,000 g, 20 min) and washed twice with ice-cold TBS (10 mM Tris-HCl pH 7.6, 150 mM NaCl). The cell pellets were frozen and stored at −20 °C.

### Chromatin immunoprecipitation for ChIP-seq

For the immunoprecipitation, cells were thawed on ice and resuspended in 360 µL of lysis buffer (10 mM Tris-HCl pH 8.0, 50 mM NaCl, 10 mM EDTA, 20% sucrose). Following lysis, 1440 µL of immunoprecipitation buffer (50 mM Tris-HCl pH 8.0, 150 mM NaCl, 1 mM EDTA, 1% Triton X-100, 0.1% sodium deoxycholate, 0.1% SDS, complete protease inhibitor cocktail) and RNaseA (up to 1 µg/mL) were added. DNA was sheared by sonication to an average size of 200 – 500 bp (17 min, pulse 10 s, pause 40 s, the amplitude 70% in Vibra-Cell VCX130 by Sonics). Cell debris was removed by centrifugation (4 °C, 13,000 × g, 15 min) and the supernatant was divided into two aliquots: 200 µL for input samples (negative control, "−IP") and 1200 µL for immunoprecipitation experiments ("+IP"). Then, 30 µL of Sera-Mag SpeedBeads Protein A/G Sepharose magnetic beads (GE Healthcare) were blocked with BSA (1 mg/mL, 15 min), and mixed with +IP supernatants, and incubated for 45 min at 4 °C to pull down proteins that non-specifically interact with the resin. Then the beads were discarded and each +IP reaction was mixed with 10 µg of purified antibodies and incubated at 4 °C overnight with rotation. The next day, the mixtures were bound to 30 µL of Sera-Mag SpeedBeads Protein A/G Sepharose magnetic beads (GE Healthcare) (blocked with BSA) by incubating for 4 h at 4 °C with rotation. Beads were washed with IP buffer, then with IP buffer containing 500 mM NaCl, and then with TE buffer (10 mM Tris-HCl pH 8, 1 mM EDTA). Cross-linked protein-DNA complexes were eluted from Protein A/G Sepharose beads with 200 µL of ChIP Elution Buffer (10 mM Tris pH 8.0, 30 mM EDTA, 1% SDS) at 65 °C for 30 min with shaking at 850 rpm. Both −IP and +IP samples were de-cross-linked by treatment with Proteinase K (Thermo Fisher Scientific) (0.2 mg/mL) at 42 °C for 2 h. The final de-cross-linking was performed at 65 °C overnight. Co-immunoprecipitated DNA was purified using the ChIP DNA Clean & Concentrator Kit (Zymo Research), according to the

manufacturer's protocol. The volume of elution was 8 µL for +IP samples and 30 µL for −IP samples. DNA concentrations were measured with a Qubit dsDNA HS (High Sensitivity) Assay Kit. There were two biological replicates of the ChIP-seq experiment.

The preparation of DNA libraries and Illumina sequencing were performed by the Skoltech Genomics Core Facility. Nine ChIP DNA libraries were prepared (for 5, 20, and 40 min time point samples) using either anti-gp64, anti-gp96 1-1540 antibodies or without the antibodies. The sequencing was performed on NextSeq platform in a paired-end mode; the length of the reads was 76 for the first biological replicate and 151 for the second biological replicate.

### ChIP-seq data analysis

The quality control of the reads was performed with FastQC (v. 0.11.5). Removing of adapters and reads with low quality was done by Skoltech Genomics Core Facility. Pre-processed reads were aligned to P23-45 phage genome (GenBank accession OP542242 where the second terminal repeat [83631... 86364] was omitted) with bowtie2 (v. 2.4.1) with default parameters for PE reads. The second terminal repeat was omitted from the phage genome since it is not possible to distinguish reads coming from identical sequences. Samtools (v. 1.10) software was used to convert.sam files to.bam format and to filter the reads. Filtering was performed using the following parameters: -F 4 (to remove all unmapped reads), -f 2 (to keep only the reads mapped in a proper pair), -q 1 (to skip alignments with mapQuality <1) (number of total and mapped reads is shown in Supplementary Data 5). MACS2 callpeak (v. 2.2.7.1) was used for the peak calling. Bam files obtained from +IP sample were passed to the program as treatment files, while those obtained from the corresponding input samples were passed as control files. MACS2 callpeak was run with the following parameters: -f BAMPE -g 83629 --keep-dup all -B. For data visualization, Integrative Genomics Viewer (IGV), v. 2.12.3 was used. Results obtained for the two biological replicates are congruent. A second biological replicate was used to create the figures.

### Sample collection and RNA purification for ONT-cappable-seq

*T. thermophilus* HB8 cells were grown in 300 mL of the liquid medium to an $OD_{600}$ of 0.2, and the culture was infected with P23-45 at a multiplicity of infection of 10. Aliquots of cells (100 mL) were collected 5 min post-infection and centrifuged for storage at −20 °C. Total RNA was purified with TRIzol (Invitrogen) according to the manufacturer's protocol. RNA pellet was dissolved in Milli-Q water, and the concentration was measured with a Qubit fluorometer. A 3 µg sample of RNA was treated with RNase-Free DNase I (Thermo Fisher Scientific) in the presence of RiboLock RNase Inhibitor (Thermo Fisher Scientific) for 30 min at 37 °C and RNA was subsequently purified by TRIzol (Invitrogen), according to the manufacturer's protocol. No glycogen was added at the isopropanol precipitation step. The pellet was dried in a SpeedVac Vacuum Concentrator (30 °C, 15 min).

### TSS and TTS determination using ONT-cappable-seq

ONT-cappable-seq was performed on an RNA sample of *T. thermophilus* 5 min post-infection with P23-45. Library preparation was performed as follows. Five µg total RNA was spiked with 1 ng of an in vitro transcribed control RNA spike-in (1.8 kb), derived from the HiScribe T7 High Yield RNA Synthesis Kit (New England Biolabs). RNA was then capped with 3′-Desthiobiotin-GTP and polyA-tailed using vaccinia capping enzyme and *E. coli* Poly(A) Polymerase (New England Biolabs), respectively. Samples were split into an enriched and non-enriched control sample before primary transcript enrichment of the former using Hydrophilic Streptavidin magnetic beads (New England Biolabs). RNA from both samples was then reverse transcribed and PCR amplified according to the PCR-cDNA barcoding protocol (SQK-PCB109; Oxford Nanopore Technologies), with an additional enrichment step using Hydrophilic Streptavidin magnetic beads after first strand

synthesis for the enriched sample. The barcoded library was pooled and loaded on a PromethION flow cell (R9.4.1; according to manufacturers' guidelines) and run on a PromethION 24 (Oxford Nanopore technologies) with live-basecalling and demultiplexing enabled. To optimize read yield, the flow cell was refueled and reloaded with the amplified cDNA library after 24 h, and the sequencing run was continued for an additional 48 h until all pores were exhausted.

The overall performance of the sequencing run and raw read quality was assessed using NanoComp (v1.11.2). Raw reads were processed and subsequently mapped to the genomes of *T. thermophilus* HB8 (NC_006461.1-NC_006463.1) and phage P23-45 (GenBank accession OP542242 where the second terminal repeat [83631... 86364] was omitted). Sequencing quality, read lengths, and mapping metrics are reported in Supplementary Data 6. It should be noted that the moderate read lengths are the result of the relatively short transcripts inherently found in the *T. thermophilus* RNA samples. This also explains the relatively low mapping percentages, as minimap2 fails to align reads shorter than ~80 bases[35]. Finally, viral transcriptional start sites (TSS) and termination sites (TTS) were determined according to the established ONT-cappable-seq workflow (Supplementary Data 1, https://github.com/LoGT-KULeuven/ONT-cappable-seq)[26]. For TSS identification, peak positions with an enrichment ratio that surpassed the stringent threshold value (TSS spike-in = 147) and had more than five absolute 5′ read end counts in the enriched samples were retained.

To discriminate between the different P23-45 RNAP promoter sequences, the regions upstream the identified TSSs (−30 to +1) were analyzed using MEME (-dna -minw 15)[36], version 5.4.0. The two resulting promoter motifs were used to conduct a FIMO (v. 5.4.0) search on the P23-45 genome to detect TSSs initially missed by the ONT-cappable-seq data analysis pipeline[37]. Based on additional motif occurrences in the viral sequence and manual inspection of the phage transcriptional landscape, seven additional phage promoters were included to the list of regulatory elements.

In parallel, for each TTS identified with ONT-cappable-seq, the adjacent −50 to +50 region was uploaded in ARNold[38] and MEME (-dna -minw 20) to predict intrinsic, factor-independent transcription termination sequences, and discover motifs, respectively[36,38].

## Cell-free activity analysis of P23-45 gp96 vRNAP

For the cell-free analysis, the truncated versions of the gp96 gene were amplified from the expression vector encoding gp96 1–1200 fragment by PCR, using the primers listed in Supplementary Table 3. Sequences of the T7 RNAP promoter and ribosome binding site were included in the forward primers. The PCR products were purified with Wizard SV Gel and PCR Clean-Up System (Promega). A volume of 0.75 μL of the DNA templates was mixed in 2.5 μL cell-free protein expression reactions (PUREFrex 2.0, GeneFrontier) and incubated at 37 °C for 1 h, and at 70 °C for 15 min to complete the protein expression and inactivate the proteins of the cell-free kit, respectively. The scaffold for the RNA extension reaction (RNA primer: 5′ FAM-CCUUGAGUCUGC GGCGAUGG; DNA template: GAGGTAGTGTGACATAGACCATC GCCGC; DNA non-template: CTATGTCACACTACCTC) was annealed in water (70 °C for 5 min, −0.1 °C/s to 12 °C). To prepare 5 μL reaction mixtures for the RNAP activity assay, the expressed gp96 fragments were mixed with the RNA/DNA scaffold (0.2 μM) and 1 mM each of ATP, CTP, GTP, and UTP. The reactions were incubated at 55 °C for 1 h and stopped by adding three volumes of the loading buffer containing 8 M urea, 20 mM EDTA, and 10 μM complement DNA to the DNA template (GCGGCGATGGTCTATGTCACACTACCTC) to remove the extended RNA primer from the DNA template. The samples were heated to 94 °C for 5 min and resolved by urea-PAGE gels (15% PA, 8 M urea). The gels were analyzed using an Amersham Typhoon Biomolecular Imager (GE Healthcare).

## Preparation, crystallization, and structure determination of the gp96 1–1200 fragment

To obtain the selenomethionine (SeMet) derivative of the 1–1200 fragment, the *E. coli* cells (BL21-Gold (DE3)) containing the expression vector were cultured in M9 medium containing SeMet[39]. The cells were disrupted by sonication, and the lysate was incubated at 65 °C for 30 min. The aggregated proteins from *E. coli* were removed by centrifugation. The gp96 1–1200 fragment was purified by chromatography on HisTrap HP, HiTrap Heparin HP, and HiLoad 16/600 Superdex 200 pg columns (GE Healthcare). Reductive methylation was further performed to improve the crystallizability of the 1–1200 fragment[40,41].

Crystallization was performed by the sitting drop vapor diffusion technique at 20 °C. Well-diffracting crystals of the 1–1200 fragment were obtained in the condition containing 20% PEG 1,000 and 100 mM Tris-HCl buffer (pH 9.0). The crystals were soaked in the crystallization solution containing 20% DMSO as a cryoprotectant, and flash-frozen in liquid nitrogen.

The X-ray diffraction data sets were collected automatically at beamline BL32XU at SPring-8 with the ZOO system[42]. The collected images were automatically processed and merged with KAMO (v. 2020-01-06)[43], using XDS (Jan 31, 2020)[44] (Supplementary Table 1).

The structure was solved by the single-wavelength anomalous diffraction (SAD) technique with PHENIX AutoSol (v. 1.17.1)[45], by using anomalous scattering from the SeMet residues. The structure was refined with the programs COOT (v. 0.8.9.2)[46] and PHENIX (v. 1.17.1)[47]. Figures were prepared using PyMOL (v. 2.5.0)[48].

## Preparation, crystallization, and structure determination of the gp96 327–1124 fragment

To construct the expression vector for the 327–1124 fragment, the truncated gene and the vector pET-47b were first amplified by PCR, by using primers InP327 (TTTCAGGGACCCGGGCGCAATAACAA-CAAACCTTTCGCCCAGCAA) and In1124 (TTCGGATCCTTATTAATC CGGGTCCTCCTT) for the gene and InF47 (CCCGGGTCCCTGAAA-GAGG) and InR47 (TAATAAGGATCCGAATTCTGTACAGGCCTTG) for the vector, respectively. Then, the amplified gene was cloned into the vector using In-Fusion (TAKARA).

The SeMet derivative of the 327–1124 fragment was expressed in the same way as the 1–1200 fragment. The cells were disrupted by sonication, and the lysate was incubated at 70 °C for 20 min. Aggregated proteins were removed by centrifugation. The 327–1124 fragment was first purified by chromatography on HiTrap Heparin HP. The expression tag was cleaved by HRV3C protease. Then, the sample was purified by chromatography on HiLoad 16/600 Superdex 200 pg columns.

Crystallization was performed by the sitting drop vapor diffusion technique at 20 °C. Well-diffracting crystals of the 327–1124 fragment were obtained in the condition containing 22.5% 2-Methyl-2,4-pentanediol, 12.5% PEG 3350, 100 mM HEPES (pH 7.5), 100 mM NH₄OAc, 100 mM MgCl₂. The crystals were flash-cooled in liquid nitrogen.

The X-ray diffraction data set was obtained at beamline NE3A of PF-AR (Supplementary Table 1) and processed with XDS (Jan 31, 2020)[44]. The structure was solved by molecular replacement with the program PHASER (v. 2.8.3)[49] using the coordinates of the corresponding region of the 1–1200 fragment as the search model. The asymmetric unit contains four 327–1124 fragments. The structure was refined with the programs COOT (v. 0.8.9.2)[46] and PHENIX (v. 1.17.1)[47]. Figures were prepared using PyMOL (v. 2.5.0)[48].

## Preparation, crystallization, and structure determination of the gp62 and gp64

For crystallization purposes, expression plasmids of each gp62 and gp64 in pET-28 vectors were transformed into the *E. coli* BL21(DE3)-Rosetta cells and grown as described above. The SeMet derivative of gp62 was grown in M9 minimal media as described for gp96. Following growth, cells were disrupted by sonication, and the lysate was clarified

by centrifugation, omitting the heat denaturation step. Each of the clarified lysates was loaded onto a 5 mL HisTrap Sepharose HP column and eluted with a linear 0–0.5 M imidazole gradient. The purest fractions were pooled and treated with thrombin to remove the polyhistidine tag. Samples were further purified by first using a 5 mL HiTrap Heparin HP column for affinity chromatography, followed by size-exclusion chromatography on a HiLoad 16/600 Superdex 200 column. The purity of the protein samples was determined using SDS-PAGE, and concentrations were quantified using a nanodrop.

Crystallization was performed by the sitting drop vapor diffusion method at 15 °C. Crystals of gp62, SeMet-gp62 and gp64 were obtained using 15-25% PEG 3350 or PEG 4000 as precipitants. Suitable crystals were harvested in the same precipitant solution supplemented with 20% glycerol before vitrification by direct immersion in liquid nitrogen. Diffraction data (Supplementary Table 2) were collected at LS-CAT (Station ID-21) at the Advanced Photon Source (Argonne, IL). Diffraction data were indexed and scaled using XDS (v. 2018)[44]. The structure of gp62 was determined using anomalous diffraction data collected on crystals of SeMet-labeled protein using PHENIX AutoSol (v. 1.16)[45]. The structures of wild-type gp62 and gp64 were determined by the molecular replacement program PHASER (v. 2.8)[49] using the coordinates of SeMet-gp62 as a search probe. The structures were built and refined with the programs COOT (v. 0.8.3)[46] and PHENIX (v. 1.16)[47]. Figures were prepared using PyMOL (v. 2.5.0)[48].

### Reporting summary

Further information on research design is available in the Nature Portfolio Reporting Summary linked to this article.

## Data availability

The ChIP-sequencing and RNA-sequencing data generated in this study have been deposited in the NCBI Gene Expression Omnibus under the GEO Series GenBank accession number GSE215812. The refined atomic models of P23-45 gp96 (327 – 1124 residues), P23-45 gp96 (1 –1200 residues), P23-45 gp64, and P74-26 gp62 generated in this study have been deposited in the Protein Data Bank under accession codes 8H2N, 8H2M, 8F72, and 8F5M, correspondingly. The unedited images of the polyacrylamide gel, in vitro transcription assays and western blot membranes can be found in 'Source Data' file. The raw mass photometry data can be found in 'Source Data Mass Photometry' file. The phage P23-45 genome used in this study is available in the GenBank database under accession code OP542242. The protein atomic models used in this study are available in the Protein Data Bank under accession codes 2J7N (N. crassa QDE1 RNAP)[25], 2O5J (T. thermophilus RNAP)[50], 6VR4 (phi14:2 RNAP)[18], 6T8H (Pyrococcus abyssi PolD DNAP)[31]. Source data are provided with this paper.

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

## Acknowledgements

The study was carried out using resources of the Skoltech Genomics Core Facility and supported by the grant from Skoltech NGP Program (Skoltech-MIT joint project) to M.L.S. S.B was supported by NIH Grant R01GM13094 and intramural funds from the Department of Cell Biology and Neuroscience at Rowan University. R.L. and L.P. were supported by the European Research Council (ERC) under the European Union's Horizon 2020 research and innovation programme [819800]. M.B. is funded by a grant from the Special Research Fund [iBOF/21/092]. S.T. was supported by JSPS (18H01328 and 22H01346) and the Japan Prize Foundation. K.S.M. and E.V.K. are funded by the Intramural Research Program of the National Institutes of Health of the USA (National Library of Medicine). Work in K.S. labs was supported by NIH RO1 grant GM59295 and Russian Science Foundation grant 19-14-00323. We thank Brovko F.A (Institute of Bioorganic Chemistry RAS, Moscow, Russia) for the immunization of rats and for providing antisera, Matvei Kolesnik (Skolkovo Institute of Science and Technology, Moscow, Russia) for reassembly of the P23-45 genome, and Petr Leiman (University of Texas Medical Branch, Galveston, US) for helpful discussions.

## Author contributions

A.C. performed biochemical experiments with gp96 and crystallized it. L.M. performed biochemical experiments with gp64. E.G. performed ChIP-seq and prepared samples for ONT-cappable-seq. B.B. and Y.H. crystalized gp64 and determined its structure. S.B. performed biochemical experiments with gp64. L.P., M.B., and R.L. performed ONT-cappable-seq. F.K. performed a mass photometry experiment. K.S.M. and E.V.K. constructed an alignment of RNAP sequences and surveyed viral genomes for the presence of two-barrel RNAPs. S.K.N. supervised the structure determination of gp64. S.T. performed biochemical experiments with gp96 and determined the structure of gp96. M.L.S. predicted TMP-fused RNAP (gp96). K.S. and M.L.S. jointly supervised the work. E.V.K., S.T., K.S., and M.L.S. wrote the manuscript, which was read, edited, and approved by all authors.

## Funding

## Competing interests

The authors declare no competing interests.
