## [Peer Review File · Nature Communications]

Tail-tape-fused virion and non-virion RNA polymerases of a thermophilic virus with an extremely long tailREVIEWER COMMENTS

Reviewer #1 (Remarks to the Author):

Chaban et al., 2023

General comments.

This is very exciting work about the structure, function and evolution of unorthodox transcription systems in the bacteriophage P23-45. The authors report on not only one but two phage-encoded RNAPs, both are of the DPBB-type but differ in very interesting ways. In particular the 'nonvirion' RNAP is fused to a virus structural protein (TMP) and the catalytic entity despite containing the two characteristic double-psi beta-barrels is highly unusual, a flat and minimal RNAP, completely lacking the RNAP clamp. Unheard of, yet functional as the authors prove experimentally! The authors document a comprehensive multidisciplinary characterisation of this dual RNAP system, encompassing not only novel X-ray structures of both enzymes, but also RNAP whole (phage) genome occupancy mapping and TSS/promoter mapping.

This is a detailed and rigorous study into the fascinating realm of phage gene expression systems, exploring the diversity of DPBB RNAPs. As such, this paper is of great of interest to the community.

Specific comments.

The article is well written and I have only a few minor suggestions for improvement.

- RNAPs typically cycle through multiple conformations as they progress through the transcription cycle, the opening/closing of the DNA binding channel, or large-scale swivelling flexibility/movements are observed in almost all DPBB RNAP. In the absence of any 3D variability analyses that cryoEM would have been provided, could the authors speculate about conformational dynamics in the two RNAPs, e. g. taking B factor analyses into account?
- The biggest challenge of rationalising the mechanisms of a clamp-less RNAP is DNA-binding during transcription elongation. How do the authors imagine the DNA binding mode of the virion RNAP? What are the DNA binding surfaces of both virion and nonvirion RNAPs? This should be included in the figure materials, if not supported by hard evidence at least the authors need to speculate about RNAP-DNA interactions.
- In their structure comparisons, it would be interesting to compare to the DPBB DNA-dependent DNAPs, archaeal polD, too.

Reviewer #2 (Remarks to the Author):

This article reports the identification and the characterization of two RNA polymerases from a T. thermophilus phage called phi 23:45.

Their putative function was inferred from the presence of the sequence motif DfDgD in their sequence.

The function was confirmed and the structure of each of them was determined by x-ray crystallography.

These structures are very interesting in that they are from a monosubunit version of the otherwise very conserved version of the multisubunit RNA Polymerases from eukaryotes, bacteria and archaea, which are typically composed of a beta subunit, a beta prime subunit and a dimer of the alpha subunit, averaging at least 400 kDa.

Here, the smallest one of the two phage RNAP is only 616 aa long, rendering it a possible common ancestor of all extant RNAP.

Actually two scenarios are possible:

-either it was transferred from a multisubunit RNAP and then pruned (simplified) to only about 620

aa, its minimum core
-or it is an ancestor of the multisubunit RNAPs.
In any case it is very interesting.

I have a few remarks below:

- p. 2 the word "implies" is too strong: "suggests"
 - p. 10 A typo line 291, extra CR
 - p. 19: there is a problem with Figure 1c: the band marked as 28n is not possible (see Fig. 1d). It looks like a +4 band, as if the RNAP has added 4 nt and then stopped. It looks like the gel has been cropped on its top part. If one compares with Figure S3b, one sees that indeed this RNAP has a tendency to make intermediate products. In any case, another gel should be produced, with an extra lane containing only the expected labelled product (28 nt), which will resolve this matter. A comment should be made about this in the text.
 - References: a key Ref. on QDE1 is missing (R. Cui et al., NAR 2022) and should be discussed in the text as well. It is clear that QDE1 is a dimer and this should be compared with the two RNAPs described here. It seems that at least the gp62 is also a dimer. Is the dimerisation mode conserved with QDE1? This is an important point that has been completely missed by the manuscript.
 - p. 36 Suppl. Table 1
 - a typo on the line CC and CC1/2
 - the lines reporting R-meas and R-merge are missing
 - the I/sigma in the last resolution shell are missing (should be in parenthesis)
 - column PDB 8H2N: presumably, the beta angle is missing for the Unit cell parameters
 - the different oligomeric states of 8H2M and 8H2N are not discussed: there is a complete contradiction between the number of atoms reported in the two different columns and the one expected by the size of the protein in the crystal?
 - Indeed 5938 atoms implies about 800 aa, as in the 327-1124 construct, while 22 384 atoms implies a dimer of 1200 aa, as in the other column: what is going on here?
 - What is the fundamental state of the protein in solution? a dimer? An AUC experiment should resolve this matter rapidly.
- p. 37: 9317 non-hydrogen atoms implies a dimer of about 620 aa. Please discuss the dimerization mode of the protein and check that it is a dimer in solution (by AUC).

Reviewer #3 (Remarks to the Author):

DNA-dependent RNA polymerases (RNAPs) are highly conserved and play critical roles in gene transcription in all three domains of life. Although the structures and catalytic mechanisms of RNAPs have been well demonstrated for some representative species, the structure and function of RNAPs from many other species are poorly understood. In this study, Chaban and coworker report the crystal structures of two RNAPs from the bacteriophage P23-45. Although the overall structure of the non-virion RNAP, gp64, is similar to many cellular RNAPs, the structure of the virion RNAP, gp96, is very unique. In combination with ChIP-seq analysis, they concluded that gp96 functions in the gene transcription at the pre-early state, whereas gp64 is responsible for the transcription of the early and middle genes. Overall, the authors present several nice crystal structures that could have interesting implications for evolution and the mechanisms for gene transcription in P23-45 and related bacteriophages. However, the story could be improved by attention to the following points:

1) As depicted in Fig. 1C, gp96_1200 and gp96_1360 have RNA primer extension abilities, whereas no activity was observed for the gp96_1540 construct. Are the oligomerization states of gp96_1200, gp96_1360 and gp96_1540 similar in solution? Are they consistent with that observed in the crystal structures? If yes, does this mean that the 1361_1540 region has an inhibitory effect on the activity of gp96?

- 2) As depicted in Fig. S4, the overall structures of gp96 RNAP 1-1200 and 327-1124 are virtually identical. How to explain the obvious difference between the catalytic activities of two constructs showed in Fig. S3b? Compared to 327-1124, the yield for the longer product is higher for the construct 327-1200. Do the extra helices at the C-terminus have any role in the activity?
- 3) In addition to DNA-dependent RNA polymerase (DdRP) activity, QDE1 and many RNAPs also possess RNA-dependent RNA polymerase (RdRP) activity. The structural basis for both the RDRP and DdRP activities has been elucidated for QDE1. Could the authors predict or test whether gp96 and gp64 also has RdRP activity?
- 4) Via ChIP-seq analysis, the authors identified many potential TSS sequence for gp96 and gp64 (Fig. 2C). Could the author use these sequences to study the de novo RNA synthesis activity of gp96 and gp64? Such activity has been demonstrated for QDE1 and many other RNAPs.
- 5) Fig. S5, Fig. S6 and Fig. S9, what are the sequences colored all in black? Please provide the detailed information in the legend.
- 6) Please check Table S2, the values for space group and cell unit were switched.

Our answers to the questions and comments below, as well as all modifications to the text and figure legends, are in blue font.

Dear Reviewers,

Thank you very much for your time and effort spent on reading, understanding, and correcting our manuscript.

REVIEWER COMMENTS

Reviewer #1

This Reviewer thought highly of our work and had “a few minor suggestions for improvement”.

- RNAPs typically cycle through multiple conformations as they progress through the transcription cycle, the opening/closing of the DNA binding channel, or large-scale swivelling flexibility/movements are observed in almost all DPBB RNAP. In the absence of any 3D variability analyses that cryoEM would have been provided, could the authors speculate about conformational dynamics in the two RNAPs, e. g. taking B factor analyses into account?

Although we attempted cryoEM analysis of the virion RNAP (gp96), the flat clamp-less RNAP had a preferred orientation in vitreous ice, which hindered the 3D reconstruction. In the two crystals of gp96 RNAP, there are five independent protein molecules in asymmetric units. All of them are in the same conformation, indicating that this conformation is a stable one. However, we now mention the possibility that the flexible, disordered part (613–679 residues) of gp96 RNAP might move dynamically during transcription to function like the clamp in multisubunit cellular enzymes (lines 185–188).

The B-factors for gp64 are fairly uniform throughout the structure and do not show any indications of conformational flexibility either. There are two copies of the protein in the crystallographic asymmetric unit, and they are superimposable.

- The biggest challenge of rationalising the mechanisms of a clamp-less RNAP is DNA-binding during transcription elongation. How do the authors imagine the DNA binding mode of the virion RNAP? What are the DNA binding surfaces of both virion and nonvirion RNAPs? This should be included in the figure materials, if not supported by hard evidence at least the authors need to speculate about RNAP-DNA interactions.

We speculate that the disordered part (613–679 residues) of the virion RNAP might interact with DNA and RNA like the clamp domain. Currently, we are preparing a series of truncation/deletion mutants of gp96 to reconstruct the minimum RNAP structure with enzymatic activity for understanding the ancient evolution of RNAPs. In preliminary experiments, the RNAP activity was lost by the deletion of this unfixed part. We consider the reconstruction of minimal RNAP as an important direction of future work, which is outside the scope of the current paper. In the revised manuscript, we added a model of the gp96 elongation complex prepared by superposition with the bacterial elongation complex (Lines 185–188, **Fig. 4**). We also changed the color of the clamp domains in **Fig. 3a** to make their positions clearer.

- In their structure comparisons, it would be interesting to compare to the DPBB DNA-dependent DNAPs, archaeal polD, too.

When we superpose P23-45 RNAPs with polD (5IJL, 6HMS), only the DPBB-A domains align well. Although DPBB-B is not modelled well in the polD structure, sequence homology between RNAPs and polD was reported for DPBB-B and the duplex-binding helix (<https://doi.org/10.1016/j.jmb.2019.05.017>). We could not find any further similarity between P23-45 RNAPs and polD. We revised the manuscript and reference to the polD papers (Lines 175–177).

Reviewer #2

This Reviewer considered our work as “very interesting” and had just “a few remarks” for us to address.

-p. 2 the word "implies" is too strong: "suggests"

Corrected

-p. 10 A typo line 291, extra CR

Corrected

-p. 19: there is a problem with Figure 1c: the band marked as 28n is not possible (see Fig. 1d). It looks like a +4 band, as if the RNAP has added 4 nt and then stopped.

It looks like the gel has been cropped on its top part. If one compares with Figure S3b, one sees that indeed this RNAP has a tendency to make intermediate products.

In any case, another gel should be produced, with an extra lane containing only the expected labelled product (28 nt), which will resolve this matter.

Fig. 1c and **Fig. 1d** differ because transcription products were resolved on polyacrylamide gels of different sizes (10x10 vs 20x20 centimeters). The difference between **Fig. 1c** and **Fig. S3** (now **Supplementary Fig. 5b**) is because different RNA/DNA scaffolds were used in these two experiments.

We repeated transcription reactions for the gp96 fragments and gp64 and resolved reaction products on identical gels along with an RNA marker and replaced **Fig. 1c** and **Fig. 1d**. Uncropped gels can be found in **Supplementary Fig. 1b**.

A comment should be made about this in the text.

We have replaced **Fig. 1c** and **Fig. 1d** and added **Supplementary Fig. 1**, showing uncropped gels.

-References: a key Ref. on QDE1 is missing (R. Cui et al., NAR 2022) and should be discussed in the text as well. It is clear that QDE1 is a dimer and this should be compared with the two RNAP described here. It seems that at least the gp62 is also a dimer. Is the dimerisation mode conserved with QDE1? This is an important point that has been completely missed by the manuscript.

Thank you. We have introduced the requested reference. To determine the oligomeric state of gp64 and gp96, we used the mass photometry technique for four variants of gp96 and gp64. All proteins

were confirmed to be monomers (Lines 91-94, **Supplementary Fig. 2**). Thus, P23-45 RNAPs in contrast to QDE-1, exist as monomers in solution.

-p. 36 Suppl. Table 1

-a typo on the line CC and CC1/2

-the lines reporting R-meas and R-merge are missing

-the I/sigma in the last resolution shell are missing (should be in parenthesis)

-column PDB 8H2N: presumably, the beta angle is missing for the Unit cell parameters

Supplementary Table 1 has been revised accordingly. R-means, R-merge, and I/sigma of the 8H2M crystal are poor as a large number of datasets were merged (Redundancy = 86.9). The high-resolution cutoff was automatically determined by the KAMO program using CC1/2, which turned out to be reasonable, resulting in pretty good Rwork/Rfree values (0.205/0.254) for a dataset with this resolution.

-the different oligomeric states of 8H2M and 8H2N are not discussed: there is a complete contradiction between the number of atoms reported in the two different columns and the one expected by the size of the protein in the crystal?

-Indeed 5938 atoms implies about 800 aa, as in the 327-1124 construct, while 22 384 atoms implies a dimer of 1200 aa, as in the other column: what is going on here?

The asymmetric units of 8H2M and 8H2N crystals contain one and four monomers, respectively, and this results in the difference in the atom numbers between the crystals that was noticed by the Reviewer. When comparing the packing interfaces of these crystals, no common interfaces were found, indicating the fundamental state of the virion RNAP is a monomer in agreement with the mass photometry experiment (Lines 91-94, **Supplementary Fig. 2**).

-What is the fundamental state of the protein in solution? a dimer? An AUC experiment should resolve this matter rapidly.

As mentioned above, we used mass photometry to demonstrate that both P23-45 RNAPs are monomers in solution (Lines 91-94, **Supplementary Fig. 2**).

-p. 37: 9317 non-hydrogen atoms implies a dimer of about 620 aa. Please discuss the dimerization mode of the protein and check that it is a dimer in solution (by AUC).

In each of the structures of gp64 and gp62, there are indeed two copies of each molecule in the crystallographic asymmetric unit. However, a comparison of crystallographic and non-crystallographic packing does not suggest any common interfaces. Therefore, these RNAPs also exist in the monomeric state in solution in agreement with the mass photometry experiment (Lines 91-94, **Supplementary Fig. 2**).

Reviewer #3

This reviewer stated that “overall, the authors present several nice crystal structures that could have interesting implications for evolution and the mechanisms for gene transcription in P23-45 and related bacteriophages”. He/she offered several points for improvement.

1) As depicted in Fig. 1C, gp96_1200 and gp96_1360 have RNA primer extension abilities, whereas no activity was observed for the gp96_1540 construct. Are the oligomerization states of gp96_1200, gp96_1360 and gp96_1540 similar in solution? Are they consistent with that observed in the crystal structures? If yes, does this mean that the 1361_1540 region has an inhibitory effect on the activity of gp96?

In the old **Fig. 1c**, the gp96_1540 fragment indeed had a weaker activity compared to gp96_1200 and gp96_1360. However, we have purified all four fragments of gp96 again in parallel for better quantitative comparison and found that while gp96_1050 was inactive, the other three fragments (1200, 1360, and 1540) had similar activity. We replaced **Fig. 1c, 1d** with new gels.

We also checked the oligomerisation state of the four gp96 fragments and gp64 by mass photometry and found that all proteins exist in solution as monomers (Lines 91-94, **Supplementary Fig. 2**).

2) As depicted in Fig. S4, the overall structures of gp96 RNAP 1-1200 and 327-1124 are virtually identical. How to explain the obvious difference between the catalytic activities of two constructs

showed in Fig. S3b? Compared to 327-1124, the yield for the longer product is higher for the construct 327-1200. Do the extra helices at the C-terminus have any role in the activity?

The cell-free activity test was done without purification or adjustment for expression levels, and thus, the result was qualitative rather than quantitative. The difference in activity between the 1-1200 and 327-1200 fragments is clear, suggesting the N-terminal region might be a self-inhibiting domain, as mentioned in the manuscript. However, we cannot be sure if the slight difference between the 327-1124 and 327-1200 fragments is meaningful. We do not think that the C-terminus of 327-1124 has an important role in the activity as it is far away from the active site.

3) In addition to DNA-dependent RNA polymerase (DdRP) activity, QDE1 and many RNAPs also possess RNA-dependent RNA polymerase (RdRP) activity. The structural basis for both the RdRP and DdRP activities has been elucidated for QDE1. Could the authors predict or test whether gp96 and gp64 also has RdRP activity?

In vitro, many RNAPs, including QDE-1, and even RNA Pol II (DOI 10.1038/nature06290), and bacterial RNAP (DOI: 10.1126/science.1134830), can synthesize RNA from both DNA and RNA templates, especially when the nature of divalent ions in the buffer is altered. Both P23-45 RNAPs extend RNA primers on RNA templates, yielding a product identical to those obtained during transcription from DNA (see **Figure X** below). Note that the major product from the scaffold used in this experiment was longer than the template and likely corresponds to RNAP slippage on the RNA template or synthesis of dsRNA through a back-priming mechanism.

Figure X. Comparison of the DNA-dependent and RNA-dependent RNA polymerization activities of gp64 and gp96. Extension of a Cy5-labeled RNA primer within a scaffold either with DNA or RNA as a template strand by gp64 and gp96_1200. Bands corresponding to the initial (10 nt) and extended (29 nt) RNA primer are labelled with one and two asterisks, correspondingly.

Presently, we do not think the RNA-dependent RNA synthesis by P23-45 RNAPs is biologically relevant since, in RNA-seq data, we do not observe any antisense RNA that could have been produced through RNA-dependent RNAP activity.

4) Via ChIP-seq analysis, the authors identified many potential TSS sequence for gp96 and gp64 (Fig. 2C). Could the author use these sequences to study the de novo RNA synthesis activity of gp96 and gp64? Such activity has been demonstrated for QDE1 and many other RNAPs.

We respectfully note that, to date, QDE-1 has not been shown to initiate RNA synthesis in a sequence-specific way. It is thought to be attracted to DNA templates by replication protein A and DNA helicase QDE-3 (<https://doi.org/10.1371/journal.pbio.1000496>). Other two-barrel RNAPs, including eukaryotic enzymes, bacterial RNAPs, and Jumbo phage RNAPs, require dedicated transcription factors/promoter specificity subunits to initiate transcription at specific sites (<https://doi.org/10.1038/s41467-022-31214-6>). We tested whether gp64 and gp96 can initiate

transcription from DNA fragments containing the predicted TSSs *in vitro*. Neither enzyme showed any activity, which likely implies that additional factors (and/or specific topological state of DNA) are required for the promoter-specific transcription initiation by these RNAPs. We now mention this in the revised text (Lines 136-138)

5) Fig. S5, Fig. S6 and Fig. S9, what are the sequences colored all in black? Please provide the detailed information in the legend.

They are long insertion sequences present in each protein. We have added the explanation in the revised manuscript (Legends for **Supplementary Figs. 7, 8, 11**).

6) Please check Table S2, the values for space group and cell unit were switched.

Apologies for the error. This has been corrected.

Additional corrections:

We revised **Supplementary Fig. 10** to correct residue numbers in the QDE-1 panel.

We added **Supplementary Fig. 1**, showing uncropped and unedited images of gels and Western blot membranes.

We introduced an **Author contributions** section.

REVIEWER COMMENTS

Reviewer #1 (Remarks to the Author):

all issues were addressed, ready to publish!

Reviewer #2 (Remarks to the Author):

The authors provided some key answers to the referees' comments.

However, I still have issues with the following points

-The authors performed mass photometry to check if the constructs were monomers or dimers. They find only monomers.

However, contrary to Ultra-centrifugation which would give the monomer-dimer equilibrium in a range of concentrations, mass photometry only gives the quaternary structure at a given concentration, in this case 10 nM, which is quite low. The authors cannot rule out a dimeric state at 0.1 or 0.5 μ M.

At the very least, this should be indicated in the text.

-The PAGE experiments reported in Fig. 1c and 1d are still not making sense to me:

The markers lane in Fig. 1c seems to indicate a product quite larger than the expected 29 nt, while Fig. 1d has the 10 nt band of the unextended primer at the same level as the one of the 17 nt of the Markers' lane.

Please do a gel with all the relevant controls in the same gel.

-In their answer to Ref. #1, (last point, comparison with polD), they mention that DPBB-B of PolD is disordered in PDB file 6HMS: this is not the latest PDB entry describing PolD: the authors should take 6T8H, at the resolution of 3.77 Angstrom, where the domain DPBB-B is well resolved.

And revise accordingly.

Reviewer #3 (Remarks to the Author):

All the concerns of the reviewer have been addressed. In the opinion of the reviewer, the manuscript is significantly improved and is acceptable for publication.

Our answers to the questions and comments below, as well as all modifications to the text and figure legends, are in blue font.

REVIEWER COMMENTS

Reviewer #2

-The authors performed mass photometry to check if the constructs were monomers or dimers. They find only monomers. However, contrary to Ultra-centrifugation which would give the monomer-dimer equilibrium in a range of concentrations, mass photometry only gives the quaternary structure at a given concentration, in this case 10 nM, which is quite low. The authors cannot rule out a dimeric state at 0.1 or 0.5 μ M. At the very least, this should be indicated in the text.

As described in the methods, for the mass photometry analysis, proteins were incubated at 2 μ M concentration for 10 min at 55 °C and then cooled to 4°C. Shortly before measurements, proteins were diluted to a final concentration of 10 nM, which took 30 seconds, and the measurements were performed for 60 seconds. All proteins were found to be monomeric in solution in these experimental conditions. We modified the methods section accordingly (Lines 361-364). Indeed, we cannot exclude the existence of dimers and any other quaternary structures under certain conditions. However, under all experimental conditions described in the manuscript, we see no signs of dimers or other oligomers, and this is explicitly stated in the manuscript.

In addition, structural considerations strongly suggest that the fundamental state of both P23-45 RNAPs is a monomer rather than a dimer. If the phage RNAPs existed as dimers at high concentrations, this would be reflected in crystal packaging as observed for homologous RNAP QDE-1 from *Neurospora crassa* (<https://doi.org/10.1371/journal.pbio.0040434>; <https://doi.org/10.1093/nar/gkac727>) and *Thielavia terrestris* (<https://doi.org/10.1074/jbc.M115.685933>). Instead, as described in the previous rebuttal letter, when comparing the packing interfaces in several crystal forms for both phage RNAPs, no common interfaces were found, indicating the monomeric state as the most probable.

-The PAGE experiments reported in Fig. 1c and 1d are still not making sense to me: The markers lane in Fig. 1c seems to indicate a product quite larger than the expected 29 nt, while Fig. 1d has the 10 nt band of the unextended primer at the same level as the one of the 17 nt of the Markers' lane. Please do a gel with all the relevant controls in the same gel.

The experiments shown in **Figure 1c** and **Figure 1d** were performed simultaneously using two identical commercial gels (Novex™ TBE-Urea Gels, 15%). As can be seen from **Supplementary Fig. 1b**, and, as expected, the unextended and extended primers have the same mobilities relative to the Marker on both gels (Merged channels, the bottom panel). The slower migration of both unextended and extended RNA primers compared to similarly sized Marker RNAs noticed by the Reviewer is due to the presence of Cy5 modification at the 5' end of the RNA primer. There is no Cy5 in the Marker RNA (it was visualized by staining with SYBR Gold). The unextended Cy5-labeled 10-nucleotide-long RNA primer migrates close to the 17-nucleotide-long RNA Marker. The extended RNA primer is 29 nucleotides long but has mobility corresponding to a 36 nucleotide-long RNA, which is an expected result since it is Cy5-labeled. Both the presence of Cy5 at the 5' end of the RNA primer and the nature of the RNA Marker are described in the legend for **Figure 1**.

-In their answer to Ref. #1, (last point, comparison with polD), they mention that DPBB-B of PolD is disordered in PDB file 6HMS: this is not the latest PDB entry describing PolD: the authors should take 6T8H, at the resolution of 3.77 Angstrom, where the domain DPBB-B is well resolved. And revise accordingly.

We thank the Reviewer for introducing the newest structure. We added the new reference and updated **Supplementary Figures 7 and 8** by adding DPBB-A and DPBB-B of PolD. We also noticed that the 'duplex-binding helix' is not helical in the latest model of PolD and thus is not conserved compared to the analyzed RNAPs. No other conserved elements were found between the analyzed RNAPs and PolD. We revised the text accordingly (Lines 177-181).